# Forecasting the Cumulative COVID-19 Cases in Indonesia Using Flower Pollination Algorithm

**Afiahayati** [1,*] , **Yap Bee Wah** [2,3,4], **Sri Hartati** [1], **Yunita Sari** [1], **I Nyoman Prayana Trisna** [5], **Diyah Utami Kusumaning Putri** [1], **Aina Musdholifah** [1] and **Retantyo Wardoyo** [1]

[1] Department of Computer Science and Electronics, Faculty of Mathematics and Natural Sciences, Universitas Gadjah Mada, Yogyakarta 55281, Indonesia
[2] Institute for Big Data Analytics and Artificial Intelligence (IBDAAI), Universiti Teknologi MARA (UiTM), Shah Alam 40450, Malaysia
[3] Centre of Statistical and Decision Sciences Studies, Faculty of Computer and Mathematical Sciences, Universiti Teknologi MARA (UiTM), Shah Alam 40450, Malaysia
[4] UNITAR Graduate School, UNITAR International University, Jalan SS6/3, SS6, Petaling Jaya 47301, Malaysia
[5] Information Technology Study Program, Faculty of Engineering, Universitas Udayana, Badung 80362, Indonesia
* Correspondence: afia@ugm.ac.id

**Abstract:** Coronavirus disease 2019 (COVID-19) was declared as a global pandemic by the World Health Organization (WHO) on 12 March 2020. Indonesia is reported to have the highest number of cases in Southeast Asia. Accurate prediction of the number of COVID-19 cases in the upcoming few days is required as one of the considerations in making decisions to provide appropriate recommendations in the process of mitigating global pandemic infectious diseases. In this research, a metaheuristics optimization algorithm, the flower pollination algorithm, is used to forecast the cumulative confirmed COVID-19 cases in Indonesia. The flower pollination algorithm is a robust and adaptive method to perform optimization for curve fitting of COVID-19 cases. The performance of the flower pollination algorithm was evaluated and compared with a machine learning method which is popular for forecasting, the recurrent neural network. A comprehensive experiment was carried out to determine the optimal hyperparameters for the flower pollination algorithm and recurrent neural network. There were 24 and 72 combinations of hyperparameters for the flower pollination algorithm and recurrent neural network, respectively. The best hyperparameters were used to develop the COVID-19 forecasting model. Experimental results showed that the flower pollination algorithm performed better than the recurrent neural network in long-term (two weeks) and short-term (one week) forecasting of COVID-19 cases. The mean absolute percentage error (MAPE) for the flower pollination algorithm model (0.38%) was much lower than that of the recurrent neural network model (5.31%) in the last iteration for long-term forecasting. Meanwhile, the MAPE for the flower pollination algorithm model (0.74%) is also lower than the recurrent neural network model (4.8%) in the last iteration for short-term forecasting of the cumulative COVID-19 cases in Indonesia. This research provides state-of-the-art results to help the process of mitigating the global pandemic of COVID-19 in Indonesia.

**Keywords:** COVID-19; forecasting; flower pollination algorithm; recurrent neural network

## 1. Introduction

COVID-19 was declared as a global pandemic by the World Health Organization (WHO) on 12 March 2020. It is an ongoing pandemic and as of 19 January 2021, more than 95.5 million cases have been confirmed, with more than 2.03 million deaths attributed to COVID-19 across 190 countries around the world [1,2]. The coronavirus was first identified in December 2019 in Wuhan, China. COVID-19 has spread globally, with America, Europe, and countries in Asia reporting high numbers of cases. The government of China quickly

implemented policies such as lockdown, physical distancing, mandatory masks, and quarantine to mitigate the spread of the virus. China has successfully controlled the pandemic rapidly and effectively, but many countries around the world are still struggling to control the spread of the virus. The virus spread to Southeast Asia on 13 January 2020, when a 61-year-old woman from Wuhan tested positive in Thailand [3]. Indonesia, a country with a population of 273 million, is the worst-hit nation in the region, with a rapid increase in cases since the first case reported in March 2020.

In the beginning, the COVID-19 pandemic has not only disrupted the normal way of life of the community, business, and government operations, but also the economy. COVID-19 has affected all levels of society and all areas of life. Hospitals and doctors are struggling to provide care for the COVID-19 patients, and businesses are affected due to lockdowns. The COVID-19 pandemic has also forced many activities to be carried out online, and new standard operating procedures (SOPs) were enforced by the government to ensure safety protocols for the public and for business operations. The COVID-19 pandemic is also causing an economic recession. The governments of many countries are allowing some economic movement while still enforcing strict health safety protocols for the public and business owners to follow. In any health disease crises, prediction of the number of cases is of utmost importance because it helps the relevant authorities to take strategic actions to mitigate the effect of the rise in numbers or control the spread of the disease.

Accurate forecasts are needed to provide useful information in the process of mitigating the global pandemic infectious disease. Thus, forecasting the number of COVID-19 cases in the upcoming few days will be most useful for considerations in making decisions, including the provision of personal equipment (PPE), preparation of economic policies, preparation of health facilities, lockdown policies, and opening of schools or businesses.

Currently, there are two approaches to forecasting COVID-19 cases. The first approach is forecasting COVID-19 using mathematical and statistical models. The mathematical and statistical model approach requires knowledge of epidemiology and statistical assumptions regarding the distribution of the data. Mathematical and statistical model approaches include the autoregressive integrated moving average (ARIMA) [4–6], seasonal ARIMA (SARIMA) [4], the susceptible-infected-recovered (SIR) model [5,7], the logistic growth model [7], and the Richards model, which is an extension of a simple logistic growth model [8].

The second approach is forecasting COVID-19 using artificial intelligence. One of the artificial intelligence approaches is machine learning. Machine learning is a computational method with sophisticated algorithms which can learn the pattern of data to solve forecasting problems. Some machine learning forecasting algorithms for forecasting COVID-19 include multi-layer perceptron, random forest, support vector regression, the Elman neural network [9–11], and the recurrent neural network (RNN) [9,10,12,13]. Sahid et al. [9] concluded that RNN outperformed support vector regression and ARIMA. Hao et al.'s [10] experimental results showed that RNN is more suitable for the prediction of the cumulative confirmed cases compared to death and cured cases.

RNN utilized network architecture which is suitable for processing sequential data. Qiu, Wang, and Zhou [14] applied RNN with long short-term memory (LSTM) architecture and attention mechanism for stock price forecasting. Uras et al. [15] applied RNN with LSTM architecture for Bitcoin closing price forecasting. Yao and Guan [16] applied RNN with an improved LSTM for natural language processing. RNN is also widely applied for speech recognition [17] and to solve fuzzy non-linear programming [18]. Hewamalage, Bergmeir, and Bandara's [19] experimental studies concluded that RNN is a good algorithm for obtaining reliable forecasts.

Another artificial intelligence approach for forecasting is a metaheuristics optimization algorithm. The flower pollination algorithm (FPA) is a robust and adaptive metaheuristics optimization algorithm which is inspired by how flower pollination occurs. The FPA solves the balance of global and local search and uses Lévy flight distribution for better global search performance. The FPA is a method that aims for optimization. The

FPA outperformed other nature-inspired methods such as the genetic algorithm and particle swarm optimization [20]. The FPA has been deployed to estimate transportation energy demand [21], to forecast Organization of the Petroleum Exporting Countries (OPEC) petroleum consumption [22], to forecast electricity energy consumption [23], and to solve combined economic and emission dispatch problems [24]. FPA was created by Yang [20] in 2014 and has been reported to perform better than other metaheuristic algorithms.

In this paper, the FPA was used to determine the optimal coefficients of the variables in the forecasting function of cumulative confirmed COVID-19 cases in Indonesia. In other words, the FPA was used to perform optimization for curve fitting of cumulative confirmed COVID-19 cases. We compare the performance of the FPA with a machine learning method which is popular for forecasting, the recurrent neural network (RNN). Experimental results showed that the FPA performed better than the RNN in long-term (two weeks) and short-term (one week) forecasting. This research provides state-of-the-art results to help the process of mitigating the global pandemic of COVID-19 in Indonesia. This paper is structured as follows: after this introduction, the second section covers related works on forecasting COVID-19 cases. This is followed by the explanation of the data and the methodology in the third and fourth section. The results and discussion are presented in the fifth section, and the conclusion is provided in the last section.

## 2. Related Works

In this section, some related works related to forecasting of COVID-19 cases are presented. As explained in the first section, there are two approaches on forecasting COVID-19 cases. The first one, the mathematical and statistical model approach, is presented here [4–8,25,26].

Mishra et al. [4] applied the ARIMA, SARIMA, and Prophet model to forecast the cumulative deaths, cumulative cases, and new cases of COVID-19 in India. The model was used to forecast the COVID-19 cases for next 15–20 days starting on 1 September 2020.

Abuhasel, Khadr, and Alquraish [5] applied SIR and ARIMA models to analyze and forecast the daily COVID-19 cases in the Kingdom of Saudi Arabia. The deterministic SIR model was applied to analyze the COVID-19 spread in Saudi Arabia, while the ARIMA model was used to forecast the daily COVID-19 cases. The two models were applied to the daily data from March 3 until 30 June 2020.

Ali et al. [6] applied the ARIMA model to forecast the cumulative confirmed cases, recovered cases, and deaths in Pakistan from COVID-19. The training data to develop the ARIMA model were from 27 February until 24 June 2020, and then the ARIMA model was used to forecast the next 10 days (25 June 2020 to 4 July 2020).

Malavika et al. [7] developed mathematical model approaches to forecast COVID-19 in India. The SIR models were applied to forecast the maximum number of active cases and peak time, the logistics growth curve model was applied for short-term prediction and the time interrupted regression model was used to analyze the effect of lockdown and other policies. The models were used to forecast the COVID-19 epidemic in India by the end of May 2020.

Zuhairoh and Rosadi [8] applied the Richards model, which is an extension of a simple logistic growth model, to forecast daily cases of COVID-19 in South Sulawesi Province, Indonesia. In addition to forecasting, the objective of this research was to predict when this pandemic would reach the peak of its spread, and when it would end. The data used in this paper were compiled as of 24 June 2020.

Anastassopoulou et al. [25] developed a mathematical model approach to estimate the fatality ratio (death rate) and recovery case ratio based on time series of positive case data, death rate, and recovered cases from COVID-19 in Hubei, China. The model was based on data distribution from Middle East respiratory syndrome (MERS) and severe acute respiratory syndrome (SARS) cases that occurred previously. The model was applied to forecast the COVID-19 cases by the end of February 2020.

Petropoulos and Makridakis [26] applied a simple time series model from the exponential smoothing family to forecast the global number of positive cases, the number of deaths, and the number of patients who have been cured of COVID-19 infection. The model was used to forecast the COVID-19 cases from February until March 2020.

The second approach in forecasting the COVID-19 cases is using artificial intelligence, especially machine learning methods [9–13]. Shahid, Zameer, and Muneeb [9] applied four different machine learning methods and the well-known ARIMA method to forecast the confirmed cases, recovered cases, and death cases in 10 major countries affected by COVID-19. The machine learning methods were RNN with bidirectional LSTM (Bi-LSTM) architecture, RNN with LSTM architecture, RNN with gated recurrent unit (GRU) architecture, and support vector regression (SVR). The data used in this research were from 22 January until 10 May 2020 for training, and from 11 May until 27 June 2020 for testing. The RNN model outperformed the SVR and ARIMA for forecasting COVID-19. The models' ranking, from the best to the worst performance, was: RNN Bi-LSTM, RNN LSTM, RNN GRU, SVR, and ARIMA.

Hao et al. [10] applied three machine learning methods to forecast the cumulative confirmed cases, cumulative deaths, and cumulative cured cases in Wuhan, Hubei Province, China. The machine learning methods were the Elman neural network, RNN-LSTM, and support vector machine (SVM). The data used in this research were from 23 January 2020 to 6 April 2020. Based on the experimental results, the RNN-LSTM model is more suitable for the prediction of the cumulative confirmed cases compared to death and cured cases.

Balli [11] applied four different machine learning time series methods to forecast the weekly cumulative confirmed COVID-19 cases for the United States of America (USA), Germany, and the world. The machine learning methods were linear regression, multi-layer perceptron, random forest, and support vector machine. The data used in this research were from between 20 January and 18 September 2020. The data consist of weekly cumulative confirmed cases for 35 weeks. SVM outperformed other methods for forecasting the COVID-19 cases.

Hawas [12] developed an RNN to forecast the data of COVID-19's daily infections in Brazil. The training data to develop the RNN model were from 7 April until 6 May 2020, and then the RNN model was used to forecast the next 54 days (7 May 2020 until 29 June 2020). In this research, there were two alternative timesteps used for the RNN, 30 and 40.

Shastri et al. [13] developed an RNN to forecast the confirmed cases and death cases of COVID-19 in India and USA. In this research, variants of LSTM architecture of RNN are developed, including stacked LSTM, bi-directional LSTM, and convolutional LSTM. The data of confirmed cases used in this research, for both India and USA, were from 7 February until 7 July 2020, while the data of death cases for India were from 12 March until July 2020, and for USA were from 26 February until 7 July 2020. The training data constituted 80% of the total, while the validation data were 20%.

In the COVID-19 research area, machine learning was used for another task beside forecasting. Machine learning has been applied to COVID-19 patient data. Zoabi et al. [27] used gradient-boosting machine model built with decision-tree base-learner for prediction of COVID-19 positive case based on symptoms while Kim et al. [28] evaluated several machine learning models to predict the need for intensive care. Recently, Ahmad et al. [29] proposed Shallow Single-Layer Perceptron Neural Network (SSLPNN) and Gaussian Process Regression (GPR) model for classification and prediction of confirmed COVID-19 cases. Elzeki et al. [30] proposed a computer-aided model using deep learning to classify positive COVID-19 based on Chest X-ray image data.

The results of closely related works are summarized in Table 1. In this research, a meta-heuristics optimization algorithm, the FPA, is used to forecast the cumulative confirmed COVID-19 cases in Indonesia. The FPA is a robust and adaptive method to perform optimization for curve fitting of COVID-19 cases. The performance of the FPA was evaluated and compared with a machine learning method which is popular for forecasting, the RNN.

**Table 1.** Summarization of closely related works.

| Authors | Methods | Forecasting of COVID-19 Cases | Results |
|---|---|---|---|
| Mishra et al. [4] | ARIMA, SARIMA, and Prophet model. | The cumulative deaths, cumulative cases, and daily confirmed cases in India. | The best root-mean-square error (RMSE) of forecasting for the cumulative cases from 23 August 2020 to 1 September 2020: 82090.21. |
| Abuhasel, Khadr, and Alquraish [5] | SIR and ARIMA models. | The daily confirmed cases in the Kingdom of Saudi Arabia. | The best RMSE of forecasting for the next 10 days: 341. |
| Ali et al. [6] | ARIMA model. | The cumulative confirmed cases, recovered cases, and deaths in Pakistan. | The best RMSE of forecasting for the cumulative confirmed cases from 25 June 2020 till 4 July 2020: 413.9. |
| Petropoulos and Makridakis [26] | A simple time series model from the exponential smoothing family. | The global number of cumulative positive cases, the number of deaths, and the number of recovered cases. | The absolute percentage error of forecasting for the cumulative confirmed cases: <br> a. 01/02/2020 till 10/02/2020: 388%; <br> b. 11/02/2020 till 20/02/2020: 7.7%; <br> c. 21/02/2020 till 01/03/2020: 6.2%; <br> d. 02/03/2020 till 11/03/2020: 12.1%. |
| Zuhairoh and Rosadi [8] | The Richards model. | The daily confirmed cases in South Sulawesi Province, Indonesia. | They provided the prediction that the peak of the COVID-19 pandemic in South Sulawesi Province, Indonesia, would be the middle of June 2020 until the end of July 2020, with 10,000–12,000 cases per day. |
| Shahid, Zameer, and Muneeb [9] | RNN with bidirectional LSTM (Bi-LSTM) architecture, RNN with LSTM architecture, RNN with GRU architecture, support vector regression, and ARIMA method. | The confirmed cases, recovered cases, and death cases in 10 major countries. | The best RMSE of forecasting for the daily confirmed cases from 11 May 2022 to 27 June 2022 (48 days): <br> a. China: 180.63; <br> b. Italy: 3612.81; <br> c. USA: 273,851.39. |
| Hao et al. [10] | Elman neural network, RNN-LSTM, and SVM. | The cumulative confirmed cases, cumulative deaths, and cumulative cured cases in Wuhan, Hubei Province, China. | The best MSE of forecasting for the cumulative confirmed cases from 24 March 2022 to 6 April 2022: 0.0320. |
| Balli [11] | Linear regression, multi-layer perceptron, random forest, and support vector machine. | The weekly cumulative confirmed cases in USA, Germany, and the world. | The best RMSE of forecasting for the weekly cumulative cases from 24 May 2022 to 18 September 2022 (17 weeks): <br> a. Germany: 329,196; <br> b. USA: 9.531,6776; <br> c. Global: 25,825.8366. |
| Hawas [12] | RNN. | The daily confirmed cases in Brazil. | R2 of forecasting for the daily confirmed cases from 7 May 2020 to 29 June 2020: 0.665. |
| Shastri et al. [13] | RNN (stacked LSTM, bi-directional LSTM, and convolutional LSTM). | The daily confirmed cases and death cases in India and USA. | The best MAPE of forecasting for the daily cases from 8 June 2020 to 7 July 2020: <br> a. India: 2.17; <br> b. USA: 2.00. |

### 3. Data

This research used cumulative daily cases data from Indonesia, which are available publicly from the Ministry of Health, Indonesia at https://kawalcovid19.id/ (accessed on 11 February 2021). Firstly, this research used data compiled since the first case reported in March 2020. This research used data from 2 March 2020, the date of the first reported case, until 24 August 2020. The data from that period are used for training and validation of models to determine the appropriate hyperparameters. After validation, the next step is testing. A detailed explanation related to the partition of training, validation, and testing data is explained in Section 4.4. The pattern of cumulative COVID-19 cases in Indonesia is presented in Figure 1.

**Figure 1.** Cumulative confirmed cases of COVID-19 in Indonesia.

## 4. Methods

### 4.1. Forecasting Using Flower Pollination Algorithm

The flower pollination algorithm (FPA) is a nature-inspired metaheuristic algorithm proposed by Yang [20]. The FPA is based on the flower pollination process of flowering plants. Flower pollination can occur by self-pollination or cross-pollination. Self-pollination refers to pollination that occurs from a different flower, or from the same flower, of a single plant. When there is no reliable pollinator available, it is usually aided by wind. Self-pollination is also referred to as abiotic pollination. Cross-pollination, on the other hand, refers to pollination from a flower of a different plant. Cross-pollination is aided by a pollinator, such as bees, bats, birds, and flies, who can fly a long distance. The pollinators may demonstrate as Lévy flight behavior. They jump or fly with distance steps that obey Lévy distribution. Cross-pollination is also referred to as biotic pollination. Cross-pollination is considered to be global pollination, while self-pollination is considered to be local pollination.

There are four rules for the FPA, based on the above flower pollination process of flowering plants:

1.  Rule 1—biotic, cross-pollination, or pollination between flowers is global pollination following Lévy Distribution. This first rule is represented mathematically in Equation (1), where $x_i^t$ is the pollen $i$ or solution vector $x_i$ at iteration $t$, $g*$ is the current best solution found among all solutions at the current iteration, and $L(\pi)$ is the strength of the pollination (step size). Lévy flight is used to mimic it; therefore, $L(\pi)$ is derived from a Lévy distribution with a value greater than 0. Lévy distribution is represented in Equation (2). Lévy distribution uses the standard gamma function $\Gamma(\pi)$, which is valid for large steps $s > 0$.

$$x_i^{t+1} = x_i^t + \gamma L(\lambda)\left(x_i^t - g*\right),\tag{1}$$

$$L \sim \frac{\lambda\Gamma(\lambda)sin(\pi\lambda/2)}{\pi}\frac{1}{s^{1+\lambda}}, (s \gg s_0 \gg 0),\tag{2}$$

2.  Rule 2—abiotic, self-pollination, or pollination of flowers from the same plants. Local pollination is represented mathematically in Equation (3). $x_j^t$ and $x_k^t$ are two pollens of the same plant but from different flowers. $\varepsilon$ is a random value from a uniform distribution in range [0,1].

$$x_i^{t+1} = x_i^t + \epsilon\left(x_j^t - x_k^t\right),\tag{3}$$

3. Rule 3–flower constancy or equivalent to a reproduction probability proportional to the likeness of the two flowers involved is often developed by the pollinators.

4. Rule 4—a probability $P \in [0,1]$ is used to switch between local pollination and global pollination.

In this study, the FPA was used to forecast cumulative cases of COVID-19. The FPA was used to obtain the best solution $g*$ from the set of solutions $x$. Each $x$ consists of a multilinear regression coefficient $\theta_l$, where $l = 1, 2, \ldots, N$ and bias $\theta_0$ to predict the cumulative daily cases of COVID-19 for day $D'_T$ based on the previous $N$ days, so that $x = \{\theta_0, \theta_1, \theta_2, \ldots, \theta_N\}$. The $\theta_l$ will be used as sum-product for $D_{T-l}$ and then the results are summed by $\theta_0$. Formally, the multilinear regression in this research is represented in Equation (4):

$$D'_T(x) = \theta_0 + \sum_{l=1}^{n} \theta_l \cdot D_{T-l}, \tag{4}$$

The objective function for each solution $x$ is to minimize the difference between predicted cumulative case $D'_T$ and actual cumulative case $D_T$. In this research, root-mean-square error (RMSE) is used to measure the difference. RMSE is presented in Equation (5), where $m$ is equal to the length of the time series record:

$$RMSE(x) = \sqrt{\sum_{i=1}^{m} \frac{\left(D'_i(x) - D_i\right)^2}{m}}, \tag{5}$$

Based on the objective function that has been determined, the fitness function for each solution to be evaluated is represented mathematically in Equation (6). The best solution for each generation is $g*$, and will be used as the final solution:

$$fitness(x) = \frac{1}{RMSE(x) + 1}, \tag{6}$$

For each generation $t$, $n$ solutions as a population are generated. From initial generation $t_0$, the best solution in the population will be stated as $g*$. In generation $t$, where $t = 1, 2, \ldots, MaxGeneration$, if there is one solution that is better than $g*$, that solution will replace the existing $g*$. The alteration of $g*$ is performed iteratively in each generation; therefore, a dynamic approach is required. The solutions in generation $t$ are formed from the pollination of the solutions in generation $t - 1$ (either global pollination or local pollination, as stated in Equations (1) and (3), respectively). The switch between global or local pollination in generation $t$ is controlled by switch probability $P$, as stated in Rule 4.

### 4.2. Forecasting Using Recurrent Neural Network

The second method applied is the recurrent neural network (RNN). RNN is a kind of neural network architecture which is suitable for processing sequential data. The advantage of the RNN architecture is that it is more flexible and can be attuned according to the number of sequences in input or output. The RNN uses iterative function cycles to store information [31]. The RNN architecture is constructed in a form such that the network will remember the previous information and apply it to calculate the current output. In the RNN, the nodes between the hidden layers are connected periodically, and the hidden layer's input includes not only the output of the input layer, but also the output of the hidden layer at the last time, thus RNN can preserve, learn, and record historical information in sequence data [32].

The RNN has a similar forward pass process to that of a multilayer perceptron with a single hidden layer. The difference lies in the fact that RNNs accept activations from both the current external input and also the hidden layer activations from previous timesteps [31]. As shown in Figure 2, the structure of the RNN includes the input layer, hidden layer, output layer, the weights of input layer to hidden layers, the weights of hidden layers

to output layers, and learnable weights for the previously hidden state. These recurrent connections serve to pass values over timestep or sequence.

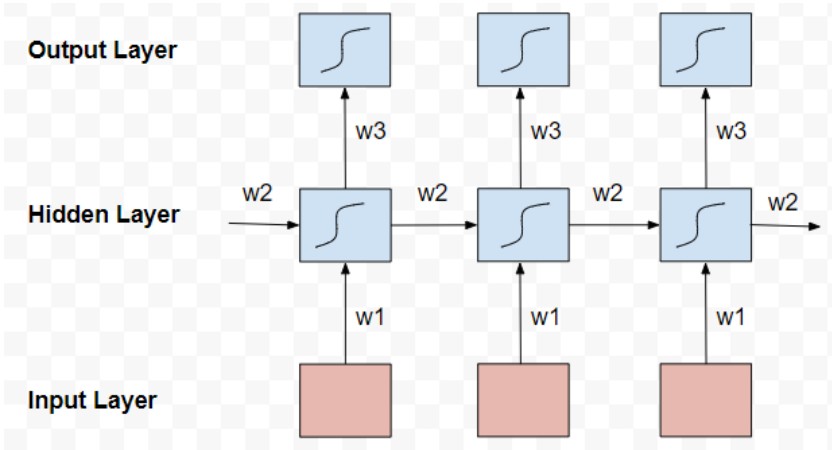

**Figure 2.** Unfolded recurrent neural network.

With this architecture, the current output in the RNN depends on the previous state. In a simple RNN, hidden units will receive the input in the current state and the output from the previous hidden state. The current hidden unit and the output can be defined mathematically in Equations (7) and (8), respectively:

$$h_t = \sigma(W_1 x_t + W_2 h_{t-1} + b), \tag{7}$$

$$o_t = W_3 h_t + b_2, \tag{8}$$

For Equation (7), $h_t$ is the hidden state and $x_t$ is the input at the current timestep. $W_1$ is the learnable weight from the input layer to the hidden layer, while $W_2$ are learnable weights for the previously hidden state's input. $\sigma$ is an activation function and $b_1$ is the bias for the hidden layer. The activation function $\sigma$ can be switched depending on the situation. The purpose of using the activation function is to ensure that the model is a non-linear machine. Common activation function choices are sigmoid, tanh, and ReLU functions. For Equation (8), $o_t$ is the output state, $h_t$ is the hidden state, $W_3$ is the learnable weight from hidden layer to the output layer, and $b_2$ is the bias for the output layer.

The complete sequence of hidden activations can be calculated by starting at the first timestep and then recursively applying Equation (7), incrementing time at each step. For the initially hidden unit at the start of the timestep, the value of the previously hidden state unit can either be manually adjusted to a certain value or set to zero. It is known that RNN stability and performance can be improved by using non-zero initial values. As for the weights, the norm is to randomize the weight without known information about the data. However, they can be set to particular values to help avoid overfitting [31].

In neural networks, the error of the prediction with respect to the target is calculated after the output is obtained. This error is normally in the form of a partial derivative of a differentiable loss function, where the derivative with respect to the weights can be used to improve the weights. There are two well-known algorithms that can be used to calculate the loss derivatives for RNNs: real-time recurrent learning (RTRL) and backpropagation through time (BPTT). BPTT is known to be simpler and more efficient in computation time, particularly since its process is similar to normal backpropagation in the neural network [31].

In this research, the data of cumulative COVID-19 daily cases are represented sequentially. Each sequence consists of data from the previous N. This sequence will be fed to RNN architectures to predict the cumulative COVID-19 cases of day $D'_T$. The experiments are conducted using several combinations of hyperparameters, such as the number of

hidden layers, the dimension of neurons, learning rates, and dropout ratio, to attempt to determine the best model with minimum RMSE. ReLU is used as the activation function and Adam is used as the optimizer.

### 4.3. Model Performance Measurement

In order to measure the performance of the forecasting model, two performance measurements are used in this research, which are root-mean-square error (RMSE) and mean absolute percentage error (MAPE). RMSE is represented mathematically in Equation (5). The smaller the RMSE values are, the more accurate the forecasting model is; conversely, the larger the RMSE values are, the more inaccurate the model is [33]. RMSE value is the error number, which doesn't provide any information about the percentage of error compared to the actual value. Meanwhile, MAPE is a widely used evaluation metric for forecasting methods presenting the percentage of error. MAPE is represented mathematically in Equation (9), where $A_t$ is actual value, $F_t$ is forecast value, and $n$ is the length of time series recorded.

$$MAPE = \frac{1}{n} \sum_{t=1}^{n} \left| \frac{A_t - F_t}{A_t} \right|, \tag{9}$$

The code of both forecasting models, the FPA and RNN, are available to be accessed publicly at http://ugm.id/covidforecasting (accessed on 29 November 2022). The code is written in Python programming language.

### 4.4. Training, Validation, and Testing Data

This study involved two phases, which are Phase 1: Development of FPA and RNN Model, and Phase 2: Evaluation of the Forecast Performance of the FPA and RNN Model Developed in Phase 1.

In Phase 1: Model Development, the data period is from 2 March 2020, to 10 July 2020. The dataset from 2 March 2020, to 10 July 2020, is divided into a ratio of 80:20; 80% for training data and 20% for validation data. Therefore, the training data are from 2 March 2020, to 4 June 2020, while the validation data are from 15 June 2020, to 10 July 2020, represented in Table 2. The validation process is carried out to determine the appropriate hyperparameters for the model.

**Table 2.** Period for developing the FPA and RNN.

| Sample | Period |
|---|---|
| Training (n = 105) | 2 March–14 June 2020 |
| Validation (n = 26) | 15 June–10 July 2020 |

In Phase 2: Model Evaluation, after the appropriate hyperparameters for the FPA and RNN model are obtained, the testing process is conducted. The FPA and RNN model is tested for short- and long-term forecast of the cumulative COVID-19 cases. We refer to some references [4–6,10,26] conducting forecasting for the next 7–14 days. Therefore, we used one-week forecast for the short-term and two-week forecast for the long-term forecasting.

1.  Long-term forecast, which forecasts the cumulative cases of COVID-19 over the next 14 days (2-week forecast);
2.  Short-term forecast, which forecasts the cumulative COVID-19 cases for the next 7 days (1-week forecast).

In order to obtain more comprehensive results of the performance of the models, the testing (forecast) process is conducted in several rounds or iterations. Long-term testing is conducted in 5 iterations, while short-term testing is conducted in 10 iterations. The model is updated with the relevant training data in each iteration using the hyperparameters defined in the validation sample in Phase 1. Table 3 presents the period of training data

and testing data for long-term testing, while Table 4 presents the period of training data and testing data for short-term testing.

**Table 3.** Period for long-term testing (forecast).

| Iteration | Types of Data | Period |
|---|---|---|
| Iteration 1 | Training Data | 2 March–15 June 2020 |
| | Testing Data | 16 June–29 June 2020 |
| Iteration 2 | Training Data | 2 March–29 June 2020 |
| | Testing Data | 30 June–13 July 2020 |
| Iteration 3 | Training Data | 2 March–13 July 2020 |
| | Testing Data | 14 July–27 July 2020 |
| Iteration 4 | Training Data | 2 March–27 July 2020 |
| | Testing Data | 28 July–10 August 2020 |
| Iteration 5 | Training Data | 2 March–10 August 2020 |
| | Testing Data | 11 August–24 August 2020 |

**Table 4.** Period for short-term testing (forecast).

| Iteration | Types of Data | Period |
|---|---|---|
| Iteration 1 | Training Data | 2 March–15 June 2020 |
| | Testing Data | 15 June–22 June 2020 |
| Iteration 2 | Training Data | 2 March–22 June 2020 |
| | Testing Data | 23 June–29 June 2020 |
| Iteration 3 | Training Data | 2 March–29 June 2020 |
| | Testing Data | 30 June–6 July 2020 |
| Iteration 4 | Training Data | 2 March–6 July 2020 |
| | Testing Data | 7 June–13 July 2020 |
| Iteration 5 | Training Data | 2 March–13 July 2020 |
| | Testing Data | 14 July–20 July 2020 |
| Iteration 6 | Training Data | 2 March–20 July 2020 |
| | Testing Data | 21 July– 27 July 2020 |
| Iteration 7 | Training Data | 2 March–27 July 2020 |
| | Testing Data | 28 July–3 August 2020 |
| Iteration 8 | Training Data | 2 March–3 August 2020 |
| | Testing Data | 4 August–10 August 2020 |
| Iteration 9 | Training Data | 2 March–10 August 2020 |
| | Testing Data | 10 August–17 August 2020 |
| Iteration 10 | Training Data | 2 March–17 August 2020 |
| | Testing Data | 18 August–24 August 2020 |

## 5. Results and Discussion

### 5.1. Hyperparameter

The validation process is conducted to obtain appropriate hyperparameters for the FPA and RNN model. The experiments engage several combinations of hyperparameters and choose the best one, providing the model with minimum RMSE. As explained in Section 4.4., the training data is from 2 March 2020, until 14 June 2020, while the validation data is from 15 June 2020, until 10 July 2020.

The combinations of hyperparameters for the FPA and RNN model are:

1. FPA Model:
    a. Length of the input timestep: 5 or 7;
    b. Switch probability between global pollination or local pollination: 0.3, 0.5, or 0.8;
    c. Population size (number of generated solutions): 50, 100, 150, or 200.
2. RNN Model:
    a. Length of the input timestep: 5 or 7;
    b. Dimension of neurons in LSTM cell: 10, 30, 50;

    c.   Learning rates: 0.001 or 0.01;
    d.   The number of hidden layers: 1 or 2;
    e.   Dropout ratio for each hidden layer: 20%, 50%, or no dropout.

In total, there are 24 combinations of hyperparameters for the FPA and 72 combinations of hyperparameters for the RNN. The best hyperparameters will be used in the testing process. For the RNN model, we use one and two hidden layers. Deeper RNN architecture required more data for training. In our research, the training data are limited enough (105 days). Therefore, if we use three or more hidden layers for the RNN, the model will have high possibility to be trapped in overfitting and it may not provide better results.

### 5.2. Results and Performance Analysis

The experiments of the validation process used 24 hyperparameter combinations for the FPA and 72 hyperparameter combinations for RNN in order to determine the best hyperparameters for this forecasting model. Based on the observation of the RMSE value for each generation, the number of generations to run the FPA is 100. The RMSE value for 100 generations reached convergence. While the number of epochs to run RNN is 1000, the RMSE value at 1000 epochs also reached convergence.

The complete 96 experiment results of the validation process are presented in Supplementary Tables S1 and S2, while the best hyperparameters, with the lowest RMSE values, are:

1.    FPA Model:

    a.   Length of the input timestep: 5;
    b.   Switch probability between global pollination or local pollination: 0.3;
    c.   Population size (number of generated solutions): 100;
    d.   RMSE value: 292.66.

2.    RNN Model:

    a.   Length of the input timestep: 7;
    b.   Dimension of neurons in LSTM cell: 10;
    c.   Learning rate: 0.01;
    d.   The number of hidden layers: 1;
    e.   Dropout ratio for each hidden layer: no dropout;
    f.   RMSE value: 502.95.

These parameters were then used to generate the FPA and RNN model for the testing process. In this validation process, the RMSE value from the FPA model (292.66) is significantly lower than that of the RNN model (502.95).

### 5.2.1. Long-Term Forecasting

There are two types of testing processes: (1) long-term forecasting for 5 iterations (different time periods); and (2) short-term forecasting for 10 iterations (different time periods). In this testing process, two performance measurements are calculated, RMSE and MAPE.

The long-term forecasting results are explained in this section, while the short-term forecasting is explained in the next section. The results for long-term forecasting are presented in Table 5. The FPA and RNN models are not overfitted, because the MAPE value for testing data is lower than the training data for all iterations. The FPA model has the lowest MAPE in the last iteration.

**Table 5.** Long-term forecasting results.

| Iteration | Data | FPA | | RNN | |
|---|---|---|---|---|---|
| | | **RMSE** | **MAPE (%)** | **RMSE** | **MAPE (%)** |
| Iteration 1 | Training Data | 185.80 | 4.12 | 177.35 | 2.57 |
| | Testing Data | 289.05 | 0.45 | 567.92 | 1.11 |
| Iteration 2 | Training Data | 394.88 | 7.74 | 760.54 | 10.91 |
| | Testing Data | 997.16 | 1.07 | 1802.61 | 2.62 |
| Iteration 3 | Training Data | 431.61 | 9.50 | 632.57 | 4.62 |
| | Testing Data | 1927.70 | 2.10 | 4639.33 | 4.93 |
| Iteration 4 | Training Data | 550.55 | 10.90 | 1082.42 | 5.03 |
| | Testing Data | 752.77 | 0.53 | 2459.13 | 2.13 |
| Iteration 5 | Training Data | 562.86 | 7.09 | 1620.60 | 6.98 |
| | Testing Data | 621.37 | 0.38 | 7715.96 | 5.31 |

Figure 3 represents a bar chart of RMSE for long-term forecasting in testing data. Figure 4 represents a bar chart of MAPE for long-term forecasting in testing data. Based on Table 5 and the clustered bar chart in Figure 3, the RMSE value of training and testing provided by the FPA model is lower than that of the RNN model for all iterations. It can be observed in Table 5 that the MAPE is higher for the FPA for the training sample at iteration 1, 3, and 4. However, the MAPE for the testing sample is lower for the FPA model compared to the RNN model, which has high MAPE values, as shown in Table 5 and Figure 3. This shows that the FPA provided more reliable long-term forecasts.

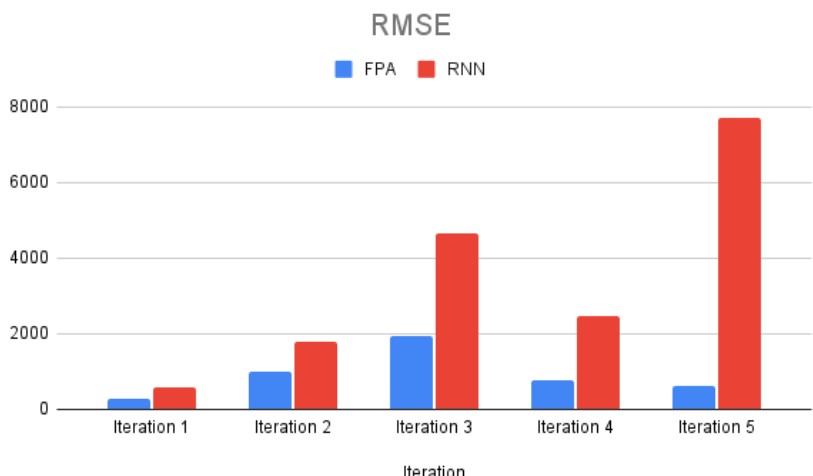

**Figure 3.** Bar chart of RMSE for long-term forecasting in testing data.

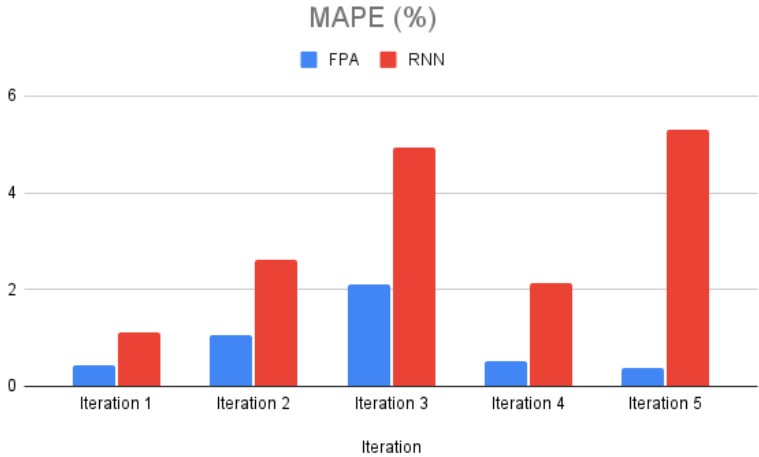

**Figure 4.** Bar chart of MAPE for long-term forecasting in testing data.

In total, until the last iteration (iteration five), we have training data consisting of 169 records (2 March 2020 until 17 August 2020). After we observed the forecasting results, it was determined that the RNN model provides more accurate forecasting in the beginning of training data (day 1–50), but less accurate in the following days. In contrast with the RNN model, the FPA model provides more accurate forecasting results than the RNN starting at day 51. The MAPE value represents the proportion between the error and the actual numbers. The RNN model provides more accurate forecasting results in the beginning, when training data contain less than 10,000 cumulative cases, but provides less accurate forecasting results in the following days, when training data contains more than 10,000 cumulative cases, reaching a total of 140,000 cases on the last day. For this reason, the RNN model has a higher RMSE value but lower MAPE value than the FPA model for iteration 3, 4, and 5 of the training data. The FPA model provides more accurate forecasting after learning, for some iterations; therefore, the FPA model provides a lower RMSE value but a higher MAPE value than the RNN model in training data. The RNN model requires more data for training. Unlike the training data, the testing data for each iteration only consists of 14 days. The next analyses will focus on forecasting results of testing data.

Figure 5 represents the trend for the actual data and long-term forecasting results using the FPA for each iteration. The x-axis represents the date and the y-axis represents the cumulative COVID-19 cases. The actual number of cumulative COVID-19 cases are represented with a blue line (real), the forecasting result for iteration 1 is represented with a red line (testing 1), iteration 2 is represented with a yellow line (testing 2), iteration 3 is represented with a green line (testing 3), iteration 4 is represented with an orange line (testing 4), and iteration 5 is represented with a black line (testing 5).

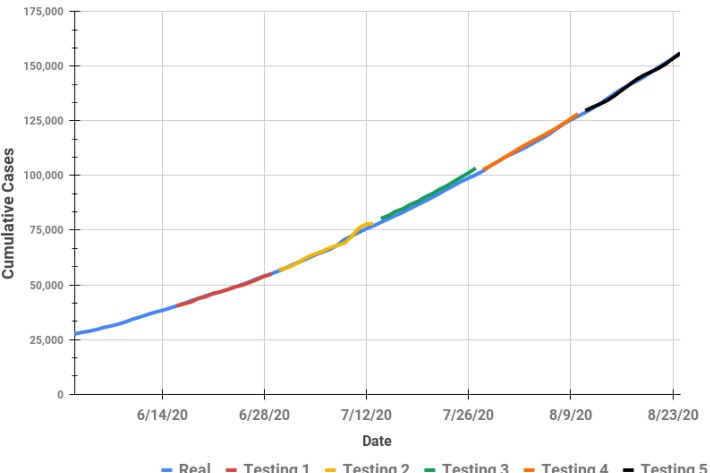

**Figure 5.** Actual and long-term forecasting for cumulative COVID-19 cases using the FPA model.

The RMSE and MAPE value for testing data in iteration 3 are the highest compared to other iterations. As we can see from Figure 5, the forecasting in iteration 3 (testing 3) learns the pattern from iteration 1 and iteration 2. The trend of data in iteration 3 has a steeper slope than the previous iterations. This may be the reason why the error value in iteration 3 is the highest one.

Figure 6 presents the trend of actual data and forecasting results for the RNN model. The forecasting results of the FPA model are better than those of the RNN model. This is also confirmed with the RMSE and MAPE results in Table 4, which shows that the overall RMSE and MAPE values of the FPA model are lower than those of the RNN model. The RNN model is a deep neural network which requires more data for training. The FPA model is better than RNN for long-term forecasting.

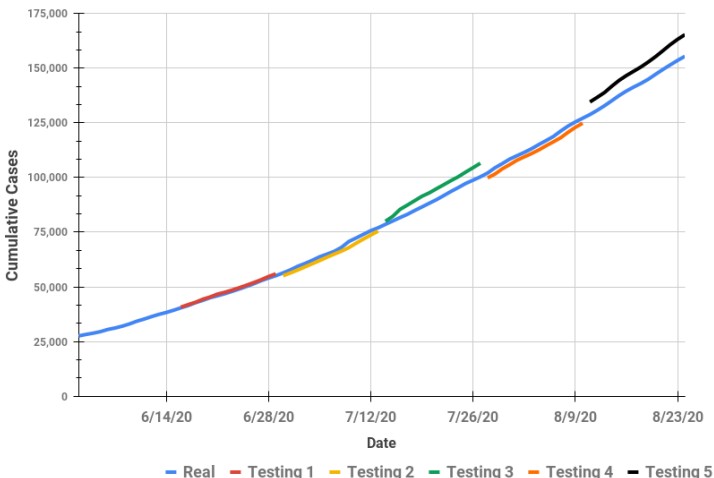

**Figure 6.** Actual and long-term forecasting for cumulative COVID-19 case using the RNN model.

5.2.2. Short-Term Forecasting

The results for short-term forecasting are presented in Table 6. The FPA model has lower RMSE for both training and testing data for all iterations except iteration 7 (RMSE for the FPA is higher than for the RNN). On the other hand, the RNN model, overall, has lower MAPE for training data, except iteration 7. The FPA has lower MAPE for testing data, except at iteration 7. These results show the RNN model may be overfitted in the iterations where MAPE is higher in testing than the training sample. Overfitting occurs when the performance of the model is good for the training but not the testing data.

**Table 6.** Short-term forecasting results.

| Iteration | Data | FPA | | RNN | |
|---|---|---|---|---|---|
| | | **RMSE** | **MAPE (%)** | **RMSE** | **MAPE (%)** |
| Iteration 1 | Training Data | 372.30 | 4.57 | 1612.35 | 3.79 |
| | Testing Data | 1179.31 | 0.74 | 7240.29 | 4.80 |
| Iteration 2 | Training Data | 167.45 | 5.49 | 582.25 | 7.92 |
| | Testing Data | 195.66 | 0.30 | 1878.50 | 3.64 |
| Iteration 3 | Training Data | 172.18 | 3.31 | 148.22 | 3.37 |
| | Testing Data | 346.66 | 0.48 | 306.01 | 0.39 |
| Iteration 4 | Training Data | 189.55 | 5.72 | 508.51 | 4.50 |
| | Testing Data | 735.18 | 0.82 | 1331.62 | 1.75 |
| Iteration 5 | Training Data | 243.84 | 5.76 | 627.81 | 2.59 |
| | Testing Data | 739.52 | 0.88 | 2467.42 | 2.89 |
| Iteration 6 | Training Data | 651.86 | 14.74 | 1031.12 | 5.51 |
| | Testing Data | 2184.86 | 2.22 | 3277.67 | 3.41 |
| Iteration 7 | Training Data | 477.98 | 10.00 | 542.96 | 5.69 |
| | Testing Data | 1589.29 | 1.34 | 221.98 | 0.13 |
| Iteration 8 | Training Data | 395.56 | 6.41 | 1166.66 | 2.91 |
| | Testing Data | 1228.22 | 0.99 | 5895.64 | 4.85 |
| Iteration 9 | Training Data | 260.75 | 3.03 | 285.52 | 2.24 |
| | Testing Data | 373.41 | 0.21 | 622.63 | 0.42 |
| Iteration 10 | Training Data | 372.30 | 4.57 | 1612.35 | 3.79 |
| | Testing Data | 1179.31 | 0.74 | 7240.29 | 4.80 |

Based on Table 6, the RMSE value of training data for iteration 4, 5, 6, 7, 8, 9, and 10 provided by the FPA model is lower than the RNN model, but the MAPE value provided by the FPA model is higher than the RNN model. This is the same as what occurred for the long-term forecasting. The RNN model provides more accurate forecasting results in the beginning, when training data contain less than 10,000 cumulative cases, but provides less accurate forecasting results in the following days, when training data contains more than

10,000 cumulative cases, reaching a total of 140,000 cases on the last day. For this reason, the RNN model has a higher RMSE value but lower MAPE value than the FPA model for iteration 4,5,6,7,8,9, and 10 in training data. The FPA model provides more accurate forecasting after learning for some iterations; therefore, the FPA model provides lower RMSE value but higher MAPE value than the RNN model in training data.

Figure 7 represents a bar chart of RMSE for short-term forecasting in testing data. Figure 8 represents a bar chart of MAPE for short-term forecasting in testing data. Figure 9 represents the trend for the actual data and short-term forecasting results of the FPA model for each iteration. The x-axis represents the date, and the y-axis represents the cumulative COVID-19 cases. The actual number of cumulative COVID-19 cases are represented with a blue line (real), the forecasting result for iteration 1 is represented with a red line (testing 1), iteration 2 is represented with a yellow line (testing 2), iteration 3 is represented with a green line (testing 3), iteration 4 is represented with an orange line (testing 4), iteration 5 is represented with a brown line (testing 5), iteration 6 is represented with a purple line (testing 6), iteration 7 is represented with a gray line (testing 7), iteration 8 is represented with a dark blue line (testing 8), iteration 9 is represented with a pink line (testing 9) and iteration 10 is represented with a black line (testing 10).

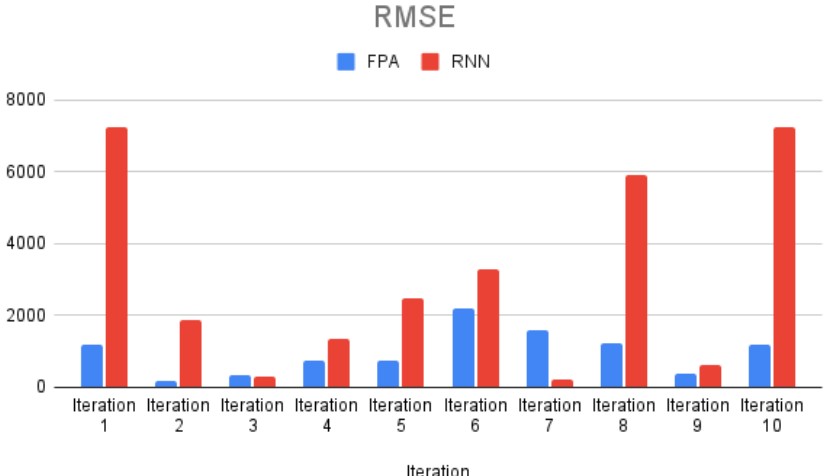

**Figure 7.** Bar chart of RMSE for short-term forecasting in testing data.

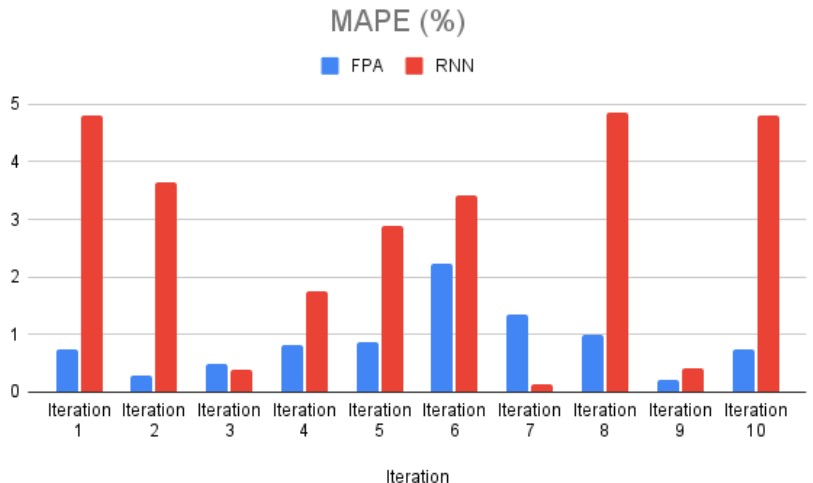

**Figure 8.** Bar chart of MAPE for short-term forecasting in testing data.

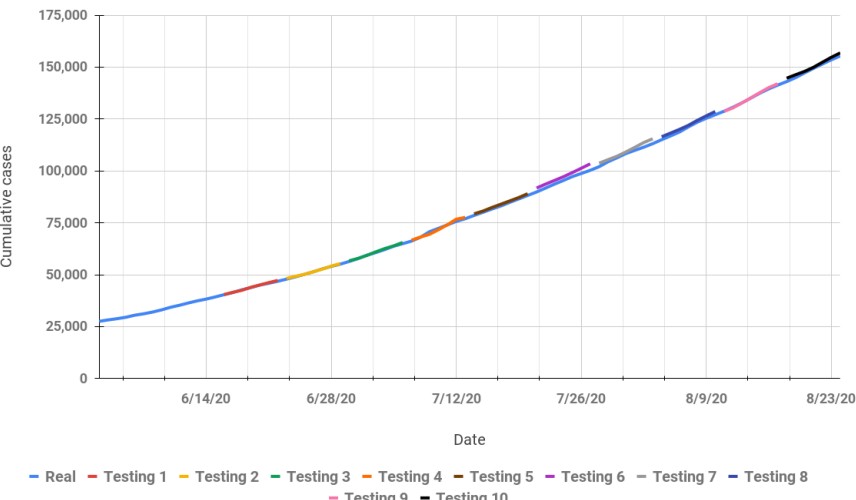

**Figure 9.** Actual and short-term forecasting for cumulative COVID-19 cases using the FPA model.

Based on Figures 7–9, as with long-term forecasting, the FPA model for short-term forecasting has the highest RMSE and MAPE value for testing data in iteration 6, which is iteration 3 for long-term forecasting. The trend of data in iteration 6 has a steeper slope than the previous iterations. The MAPE value for testing data in iteration 7 provided by the FPA model (1.34 %) is higher than that of the RNN model (0.13%). The FPA model learned a new pattern of data in iteration 6, with a steeper slope; therefore, the FPA model has the highest MAPE in iteration 6 (2.22%). The MAPE decreases in iteration 7 (1.34%) and in the next iterations. This does not occur in long-term forecasting. The FPA model can learn a new pattern of data better in long-term forecasting, which is the training data updated for 2 weeks.

Figure 10 represents the trend of actual data and forecasting results for the RNN model. Based on Figure 8, the highest MAPE value is in iteration 8, which is confirmed with the forecasting result in Figure 10 (testing 8). The forecasting results of the RNN model for the long-term model are better than for the short-term model. The training model in the RNN may not be adequately up to date with the addition of 1 week of data for each iteration. The RNN could not calculate the pattern of the data with the addition of only a few data (1 week of data).

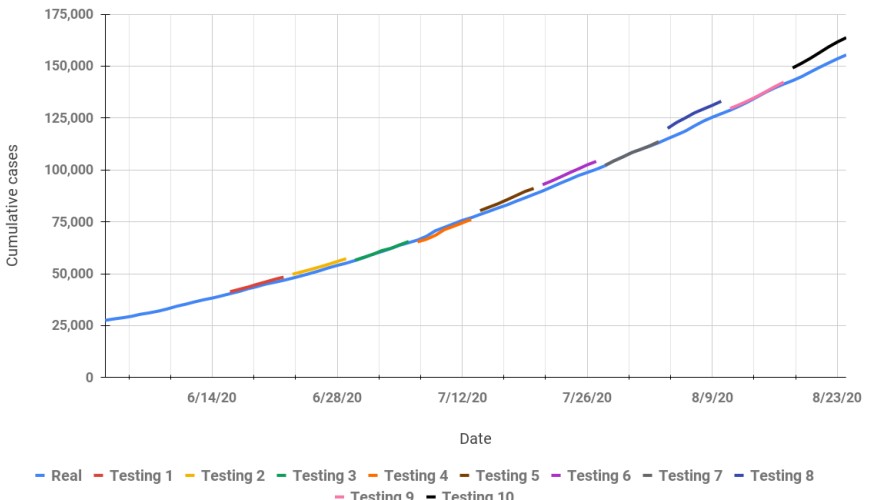

**Figure 10.** Actual and short-term forecasting for cumulative COVID-19 cases using the RNN model.

Overall, the forecasting results of the FPA model are better than the RNN model, both for long-term forecasting and short-term forecasting. The FPA model is better than the

RNN model in the presence of limited training data. The RNN model requires more data for training and to learn the pattern of data. The FPA model is better than the RNN model for forecasting the cumulative COVID-19 cases in Indonesia.

## 6. Conclusions

In this research, we presented forecasts of the cumulative COVID-19 cases in Indonesia using the FPA, a natured-inspired algorithm, to determine the optimal coefficients of the variables in the forecasting function of COVID-19 cases. We compared the performance of the FPA with a machine learning method which is popular for forecasting, the RNN. Several comprehensive experiments were conducted to determine the optimal hyperparameters for the FPA and RNN. The best hyperparameters were used to develop a model for forecasting. Long-term and short-term forecasting were conducted using different iterations with data added as more cases were reported. The FPA model has lower MAPE value than the RNN model for both long-term and short-term forecasting. These results show that the FPA model is better than the RNN model for forecasting cumulative COVID-19 cases. The FPA model was able to provide more reliable forecasts. This research provides state-of-the-art results to aid the process of mitigating the global pandemic of COVID-19 in Indonesia. In future, this forecasting model will be extended for COVID-19 active cases and deaths. Then, the forecasting results will be provided online and updated each day by developing an online dashboard for users; therefore, it will be more useful.

**Supplementary Materials:** The following supporting information can be downloaded at: https://www.mdpi.com/article/10.3390/computation10120214/s1, Table S1: The experiment results of validation process using 72 hyperparameter combinations for RNN.; Table S2: The experiment results of validation process using 24 hyperparameter combinations for the FPA.

**Author Contributions:** Conceptualization, A.; Methodology, A., Y.S., I.N.P.T. and R.W.; software, A. and I.N.P.T.; validation, A. and I.N.P.T.; formal analysis, A., Y.B.W., S.H., Y.S., A.M. and R.W.; investigation, Y.B.W. and A.M.; data curation, D.U.K.P.; writing—original draft, A., S.H., Y.S. and D.U.K.P.; writing—review and editing, A., Y.B.W. and S.H.; visualization, A.; supervision, S.H.; project administration, A.; funding acquisition, A. All authors have read and agreed to the published version of the manuscript.

**Funding:** This work is supported in part by the Department of Computer Science and Electronics, Faculty of Mathematics and Natural Sciences, Universitas Gadjah Mada, Schema C Research Grant No. 224/J01.1.28/PL.06.02/2020.

**Data Availability Statement:** The dataset and the code of the forecasting model are available to be accessed publicly at http://ugm.id/covidforecasting (accessed on 29 November 2022).

**Acknowledgments:** This work was carried out under the scheme of research collaboration between Intelligent System Laboratory, Department of Computer Science and Electronics, Universitas Gadjah Mada (UGM), Indonesia and Advanced Analytics Engineering Centre (AAEC), Faculty of Computer and Mathematical Sciences, Universiti Teknologi MARA (UiTM), Malaysia. We would like to thank Nurbaizura Borhan and Puan Aida Wati Zainan Abidin from UiTM, Malaysia, and Ilona Usuman, Anifuddin Azis, and Sri Mulyana from UGM for providing their suggestions regarding this research.

**Conflicts of Interest:** The authors declare no conflict of interest.

## Abbreviations

The following abbreviations are used in this manuscript:

| | |
|---|---|
| ARIMA | Autoregressive integrated moving average |
| Bi-LSTM | Bidirectional LSTM |
| BPTT | Backpropagation through time |
| COVID-19 | Coronavirus disease 2019 |
| FPA | Flower pollination algorithm |
| GRU | Gated recurrent unit |
| LSTM | Long short-term memory |

| MAPE | Mean absolute percentage error |
| MERS | Middle East respiratory syndrome |
| OPEC | Organization of the Petroleum Exporting Countries |
| PPE | Personal equipment |
| ReLU | Rectified linear activation function |
| RMSE | Root-mean-square error |
| RNN | Recurrent neural network |
| RTRL | Real-time recurrent learning |
| SARIMA | Seasonal ARIMA |
| SARS | Severe acute respiratory syndrome |
| SIR | Susceptible-infected-recovered |
| SOP | Standard operating procedures |
| SVM | Support vector machine |
| SVR | Support vector regression |
| USA | United States of America |
| WHO | World Health Organization |

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
