# Peer review of "Forecasting the Cumulative COVID-19 Cases in Indonesia Using Flower Pollination Algorithm"

_computation, doi:10.3390/computation10120214_

Round 1

Reviewer 1 Report

The manuscript is interesting and well-written. The methods are well explained in a manner that makes the research reproducible. The language is clear and understandable. Overall, the manuscript is a pleasure to read. There are only a few minor suggestions that I can provide regarding it:

1. It would be beneficial to the study if the results of the related research were presented in a table form, to help with comparison of the achieved research with state of the art. This could be added at the end of the section 2.
2. Please add a reference or a link to the dataset (section 3, line 190).
3. I don't see a point in presenting the results achieved during training in figures 3, 4, 7 and 8. It is expected for the training results to be good, while the test results are the ones better indicative of the performance of the developed models. As it is presented now, it makes the results hard to follow. I would suggest that authors do one of the three things, whichever they think is the best fit for their manuscript: a) completely remove the training results, b) separate training and testing results into separate graphs, c) leave the figures as-is and provide a more in-depth explanation of why they consider that both should be present in the manuscript text.

As such, I reccommend the publication of the manuscript after minor revisions are performed.

Kindest regards,
Reviewer

Reviewer 2 Report

The paper with title (Forecasting The Cumulative COVID-19 Cases in Indonesia using Flower Pollination Algorithm).

The paper is nice and can be publish after the following comments:

1- The abstract should be rewritten without abbreviations.

2- List of abbreviation should be add at the end of the paper.

3- All equations should be ended by , or.

4-All refs should be write by the same way.

5- Future directions should be add.

6- English should be review by native person because there are many errors.

Reviewer 3 Report

Afiahayati et al. used a metaheuristics optimization algorithm to forecast the cumulative confirmed COVID-19 cases in Indonesia. The manuscript is well-written and the authors presented exciting results. I am glad to see the details of the method are well documented. I only have a few comments. I recommend a minor revision of this manuscript. 

  1. Line 107: It is interesting to see that authors used 2-week as long-term forecasting. Why 2-week? Have authors tried more extended periods for their tests?

  2. Line 365: Why not 3 layers for RNN? 

  3. Is that possible to use a trained model from one country and predict for another? What are the limitations or caveats? 

  4. Is there a section for code availability in the manuscript?
